# ADVERSARIAL ROBUSTNESS OVERESTIMATION AND INSTABILITY IN TRADES

## ABSTRACT

This paper examines the phenomenon of probabilistic robustness overestimation in TRADES, a prominent adversarial training method. Our study reveals that TRADES sometimes yields disproportionately high PGD validation accuracy compared to the AutoAttack testing accuracy in the multiclass classification task. This discrepancy highlights a significant overestimation of robustness for these instances, potentially linked to gradient masking. We further analyze the parameters contributing to unstable models that lead to overestimation. Our findings indicate that smaller batch sizes, lower beta values (which control the weight of the robust loss term in TRADES), larger learning rate, and higher class complexity (e.g., CIFAR-100 versus CIFAR-10) are associated with an increased likelihood of robustness overestimation. By examining metrics such as the First-Order Stationary Condition (FOSC), inner-maximization, and gradient information, we identify the underlying cause of this phenomenon as gradient masking and provide insights into it. Furthermore, our experiments show that certain unstable training instances may return to a state without robust overestimation, inspiring our attempts at a solution. In addition to adjusting parameter settings to reduce instability or retraining when overestimation occurs, we recommend incorporating Gaussian noise in inputs when the FOSC score exceed the threshold. This method aims to mitigate robustness overestimation of TRADES and other similar methods at its source, ensuring more reliable representation of adversarial robustness during evaluation.

## 1 INTRODUCTION

Adversarial robustness has emerged as a critical measure of security in machine learning models, particularly for deep neural networks. These models, while achieving state-of-the-art accuracy on various tasks, often exhibit vulnerabilities to adversarial examples—inputs specifically crafted to cause the model to make errors (Szegedy et al., 2014). Adversarial training (Goodfellow et al., 2015; Madry et al., 2018), a method that involves training a model on adversarial examples, has been developed as a primary defense mechanism to enhance adversarial robustness.

In a more recent development, Zhang et al. (2019) proposed a novel adversarial training method, TRADES (TRadeoff-inspired Adversarial DEfense via Surrogate-loss minimization). Consider a data distribution $D$ over pairs of examples $\mathbf{x} \in \mathbb{R}^d$ and labels $y \in [k]$. As usual, $f_\theta$ represents the classifier, $\mathcal{CE}$ is the cross-entropy (CE) loss, and $\mathcal{KL}$ is the Kullback-Leibler (KL) divergence. For a set of allowed perturbations $S \subseteq \mathbb{R}^d$ that formalizes the manipulative power of the adversary, TRADES has the loss minimization defined as:

$$\min_f \mathbb{E}_{(\mathbf{x},y)\sim D}\left\{\mathcal{CE}(f(\mathbf{x}), y) + \frac{1}{\lambda}\max_{\delta \in S}\mathcal{KL}(f_\theta(\mathbf{x}), f_\theta(\mathbf{x} + \delta))\right\}. \tag{1}$$

Within this approach, the inner maximization seeks to increase the KL-divergence between the logits generated from the original and adversarial examples, while the outer minimization aims to balance accuracy and robustness by utilizing a hyperparameter $\lambda$ and adjusting the trade-off between the performance on natural and adversarial examples. This approach has been successful and is commonly used today as one of the bases for adversarial training, often serving as a baseline.

However, the loss design of TRADES is similar to the logit pairing method (Kannan et al., 2018), which is an abandoned approach due to its potential to cause gradient masking (Athalye et al., 2018)

(a typical flaw in defense methods that appear to fool gradient-based attacks but do not actually make the model more robust). In the past, many analyses (Mosbach et al., 2018; Engstrom et al., 2018; Lee et al., 2021) have been conducted on this issue, but none of the works have experimented on TRADES specifically. As a result, TRADES is still trusted and widely used today.

Through our study, under certain hyperparameter settings in multi-class classification tasks, we found that some training instances may lead to robustness overestimation, where AutoAttack (Croce & Hein, 2020b) test accuracy is significantly lower than PGD-10 (Madry et al., 2018) validation accuracy. After analyzing the loss landscapes and the convergence capability of multi-step adversarial examples of these instances, we observed the occurrence of gradient masking, a phenomenon where the adversarial robustness of a model is inaccurately gauged, potentially compromising the effectiveness of the defense in practical scenarios.

Moreover, by examining the inner maximization and batch-level gradient information, we experimentally explained the reasons behind this phenomenon. We also found that robustness overestimation may occasionally end by the model itself, returning to stability —a phenomenon we refer to as "self-healing". Inspired by some of the characteristics of self-healing, we proposed an actively healing solution by simultaneously quantifying the degree of gradient masking and adding Gaussian noise in inputs if needed, which allows errors during training to be detected and corrected in real-time without the need for retraining. The contributions of this paper are as follows.

1. We identify the adversarial robustness overestimation and instability in TRADES and the impact of hyperparameters in multi-class classification. By examining the loss landscape and the convergence capability of adversarial examples, we interpret this issue as gradient masking.

2. We analyze inner-maximization metrics and gradient information to elucidate the underlying causes of the observed phenomena and propose methodologies to address these instabilities. Additionally, we discover that some training instances have the potential to self-heal and resolve the issues mentioned above.

3. We propose a healing solution that enables real-time examination and correction of potential instability during TRADES training.

Our findings aim to refine the understanding and application of current adversarial training frameworks, paving the way for more secure machine learning deployments.

## 2 RELATED WORK

To defend against various adversarial attacks (Carlini & Wagner, 2017; Madry et al., 2018; Croce & Hein, 2020a; Andriushchenko et al., 2020; Croce & Hein, 2020b), adversarial training (Goodfellow et al., 2015; Madry et al., 2018) has been demonstrated to be effective in enhancing the robustness with AutoAttack (Croce & Hein, 2020b) serving as a reliable evaluation method. (Croce et al., 2021) Adversarial training comes in different variants, including PGD-AT (Madry et al., 2018), ALP (Kannan et al., 2018), and LSQ (Shafahi et al., 2019), MMA (Ding et al., 2020), among others. On the other hand, Tsipras et al. (2019); Zhang et al. (2019) have identified a trade-off between robustness and accuracy, a phenomenon well-explained in theoretical aspects. Based on this, Zhang et al. (2019) developed TRADES, which currently serves as a classical baseline in adversarial training. Then, subsequent works include Wang et al. (2020); Zhang et al. (2020); Blum et al. (2020); Jin et al. (2022; 2023) have improved upon TRADES and achieved promising results.

However, in adversarial training, logit pairing methods (such as the previously mentioned ALP (Kannan et al., 2018) and LSQ (Shafahi et al., 2019)) have been analyzed as unreliable because they lead to robustness overestimation. (Mosbach et al., 2018; Engstrom et al., 2018; Lee et al., 2021) This typically arises from gradient masking (or obfuscated gradients) (Athalye et al., 2018; Goodfellow, 2018; Boenisch et al., 2021; B.S. et al., 2019; Ma et al., 2023), which results in black-box attacks performing unexpectedly better than white-box attacks. This is unacceptable because the model does not provide genuinely reliable robustness. Therefore, for a more comprehensive robustness evaluation, in addition to using different attacks for testing, directly quantifying (Wang et al., 2019; Liu et al., 2020; Lee et al., 2021) or visualizing (Li et al., 2018; Liu et al., 2020) the relationship between the testing model and adversarial examples is also a good approach.

## 3 EXPERIMENTAL METHODS AND SETTINGS

### 3.1 RELEVANT HYPERPARMETERS

We identify several key hyperparameters that impact instability, which will be discussed in detail in Section 4.1. These are listed here in advance: (1) $\beta$ (referred to as $\frac{1}{\lambda}$ in Equation 1), (2) batch size, and (3) learning rate. To better analyze the phenomenon, unless otherwise specified, our default setting is $\beta = 3$, batch size = 256, and learning rate = 0.1. Also, the complexity of the dataset may also influence the results, with CIFAR-100 being used as the default. (The above setting is relatively reasonable and similar to (Zhang et al., 2019; Pang et al., 2021; Wu et al., 2024)) In addition, other experimental settings are provided in Appendix B.

### 3.2 EVALUATION METHODOS

**Adversarial Attacks** To measure the model's robustness, the most direct approach is to use adversarial attacks to test each sample and obtain the robust accuracy. Generally, Projected Gradient Descent (PGD) (Madry et al., 2018), a well-known and highly effective white-box attack, is frequently used for validation to select checkpoints. Additionally, for more reliable robustness evaluation, AutoAttack (Croce & Hein, 2020b) has become the mainstream benchmark method. It combines four different attacks: APGD-CE (Croce & Hein, 2020b), APGD-DLR (Croce & Hein, 2020b), FAB (Croce & Hein, 2020a), and Square Attack (Andriushchenko et al., 2020). Among these, Square Attack is of particular interest, as it is a black-box attack that is highly relevant in identifying the presence of gradient masking.

**First-Order Stationary Condition (FOSC)** To measure the degree of gradient masking, we utilize FOSC (Wang et al., 2019) to assess the convergence capability of multi-step adversarial examples. Suppose we have a k-step adversarial example $\mathbf{x}^k$, its FOCS is defined as:

$$FOSC(\mathbf{x}^k) = \max_{\delta \in S} \left\langle \mathbf{x} + \delta - \mathbf{x}^k, \nabla_x \ell(\mathbf{x}^k) \right\rangle, \tag{2}$$

where $\ell(\mathbf{x}^k) = \mathcal{CE}(f_\theta(\mathbf{x}), y)$ denotes the loss of the attacker. A smaller FOSC value indicates stronger attack convergence capability, hence a lower level of gradient masking.

**Step-wise Gradient Cosine Similarity (SGCS)** Another metric we refer to is SGCS Lee et al. (2021), which can also be used to compare the convergence stability. For the same k-step adversarial example $\mathbf{x}^k$, let $\mathbf{x}^i$ be the resultant example from the $i$-th step of the attack, we have:

$$\mathbb{K} = \{(i, j) \mid i, j \in \{0, 1, \ldots, k-1\}, i \neq j\},$$

$$g(\mathbf{x}^i) = \text{sign}(\nabla_{\mathbf{x}^i} \ell(\mathbf{x}^i)), \tag{3}$$

$$SGCS(\mathbf{x}^k) = \mathbb{E}_{(\mathbf{x},y) \sim D} \left[ \frac{1}{k(k-1)} \sum_{(i,j) \in \mathbb{K}} CosSim(g(\mathbf{x}^i), g(\mathbf{x}^j)) \right],$$

where $CosSim(g(\mathbf{x}^i), g(\mathbf{x}^j))$ denotes the cosine similarity of the adversarial example at different steps. SGCS is the alignment of gradients for a multi-step adversarial example, which indicates a higher degree of gradient masking if the value is smaller.

## 4 IDENTIFYING ROBUSTNESS OVERESTIMATION AND GRADIENT MASKING

### 4.1 IDENTIFYING ROBUSTNESS OVERESTIMATION

Motivated by the need to assess the reliability of TRADES, we conduct robust evaluations using different attacks. However, we discover that, under identical configurations, two distinct instances initialized with different random seeds can exhibit significant performance variation. We use PGD-10 during validation to select the best checkpoint and perform a more rigorous evaluation at test time using the more reliable AutoAttack. The results (Table 1) show that robustness overestimation probabilistically occurs in certain instances, where the PGD-10 validation accuracy is significantly

|          | Clean  | PGD-10     | AutoAttack | APGD-CE | APGD-DLR | FAB    | Square     |
|----------|--------|------------|------------|---------|----------|--------|------------|
| Regular  | 0.5363 | 0.2441     | 0.2027     | 0.2393  | 0.2128   | 0.2167 | 0.2426     |
| Unstable | 0.5593 | **0.2872** | **0.1042** | 0.1924  | 0.1885   | 0.1920 | **0.1215** |

Table 1: Under the same default configuration (CIFAR-100, $\beta = 3$, batch size = 256, learning rate = 0.1), with different random seeds, some training instances lead to robustness overestimation, where PGD-10 accuracy is significantly higher than the more reliable AutoAttack. We refer to these instances as "Unstable" cases, in contrast to the "Regular" cases, which exhibit stable and expected behavior.

| Learning Rate = 0.1 | | | | Batch Size = 128 | | | |
|---|---|---|---|---|---|---|---|
| $\beta$ \ batchsize | 128 | 256 | 512 | $\beta$ \ Learning Rate | 0.01 | 0.05 | 0.1 |
| 1 | **10**/10 | **10**/10 | **10**/10 | 1 | **10**/10 | **10**/10 | **10**/10 |
| 3 | **10**/10 | **7**/10 | 0/10 | 3 | 0/10 | **10**/10 | **10**/10 |
| 6 | 1/10 | 0/10 | 0/10 | 6 | 0/10 | 1/10 | 1/10 |

Table 2: Probability of unstable cases w.r.t. the influential hyperparameters. For each set of parameters, we perform training 10 times with random seeds to identify groups that may exhibit gradient masking, which are identified when the PGD-10 (white-box) accuracy is significantly higher than the Square Attack (black-box) accuracy.($\geq 8\%$).

higher than the AutoAttack test accuracy. The main contributor to this discrepancy is Square Attack within AutoAttack, as we find that the black-box attack (Square Attack) outperforms the white-box attack (PGD), indicating **the presence of gradient masking as the primary cause of the anomaly**. For future reference, we refer to such anomalous cases as "Unstable" cases, highlighting that TRADES may unexpectedly induce gradient masking. Analyzing the causes and resolving this issue is the primary focus of this work.

In the preceding empirical observations, we identify that robustness overestimation is not uniformly present across all training instances, even when hyperparameters remain constant. This variability highlights the need for a deeper examination of the factors contributing to this phenomenon's instability. Therefore, we begin by exploring the hyperparameters (Table 2)that may influence this instability and find that $\beta$ (the coefficient regulating the trade-off between natural and adversarial loss), batch size, and learning rate all have a significant impact. We discover that smaller values of $\beta$, smaller batch sizes, larger learning rates, and higher class complexity of the datasets (see Appendix C) exhibit a positive correlation with increased instability. In addition, the unstable cases can still occur when applying learning rate scheduling (see Appendix D).

## 4.2 Exploring the Instability of Gradient Masking

To more clearly depict gradient masking in unstable cases, we plot the data loss landscapes in Figure 1 from the corresponding checkpoints. Typically, more robust models tend to have smoother loss surfaces, whereas less robust models exhibit more jagged landscapes (Chen et al., 2021). We find that checkpoints with an overestimated robust accuracy tend to correspond to these jagged loss landscapes, which supports that unstable cases obtain less robust representations, thereby leading to gradient masking.

We support our perspective by utilizing FOSC (Wang et al., 2019) and SGCS (Lee et al., 2021) (described in Section 3.2). These metrics represent values that measure the degree of adversarial convergence and can be used to assess the ruggedness of the loss landscape. As depicted in Figure 2, one can observe that the difference in PGD-10 accuracy between the two cases is not significant. However, both FOSC and SGCS in the unstable case "simultaneously" show a sharp change, indicating a substantial loss of robustness and the occurrence of gradient masking, which results in the PGD attacker struggling to find adversarial examples. We also provide a theoretical explanation of the relationship between FOSC and SGCS in Appendix E.

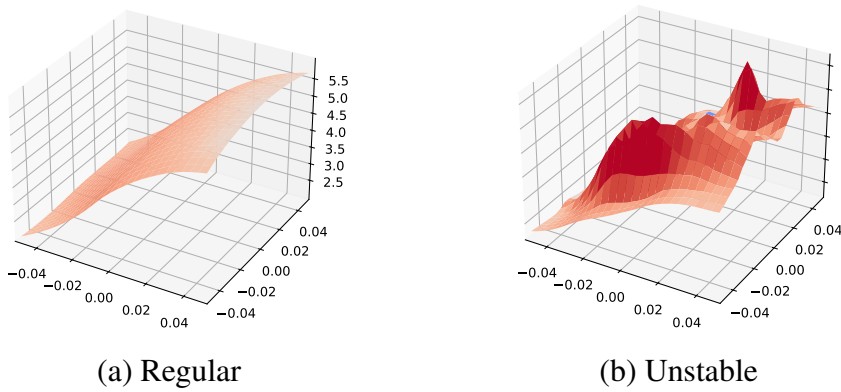

(a) Regular                                          (b) Unstable

Figure 1: Comparison of the data loss landscapes for regular and unstable cases under the same configuration. Following the same setting of Wu et al. (2024); Engstrom et al. (2018); Chen et al. (2021), we plot the loss landscape function $z = \text{loss}(x \cdot r_1 + y \cdot r_2)$, where $r_1 = \text{sign}(\nabla_i f(i))$ ($i$ is the input data) and $r_2 \sim \text{Rademacher}(0.5)$. The $x$ and $y$ axes represent the magnitude of the perturbation added in each direction and the $z$ axis represents the loss. One can observe that the loss landscape of the unstable case is highly rugged, which is not expected for a robust model.

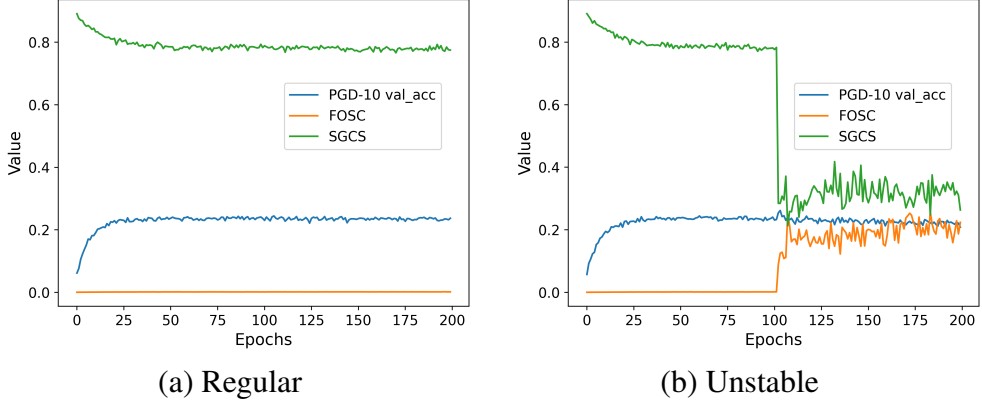

(a) Regular                                          (b) Unstable

Figure 2: Comparison between the FOSC, SGCS, and PGD-10 validation accuracy. Under the same configuration but with different seeds, the values for the regular case are displayed in (a), while the unstable case is shown in (b). Note that for clearer visualization, we scaled FOSC up by 10. Although the PGD-10 validation accuracy shows only slight fluctuations, both FOSC and SGCS exhibit significant changes within the same epoch, indicating that we can indeed observe gradient masking through this relation.

## 5 ANALYZING AND INTERPRETING THE INSTABILITY

In this section, we analyze robustness overestimation and instability from three different perspectives: 1) inner maximization, 2) gradient information, and 3) self-healing. Through this analysis, we aim to understand the causes of instability and explore potential solutions. Also, from this point onward, we use the same seed of unstable instance as default settings in Figure 2 to better explain the instability (though this does not imply that these explanations are incidental). Additionally, we now focus on FOSC as a more effective tool for detecting unstable cases.

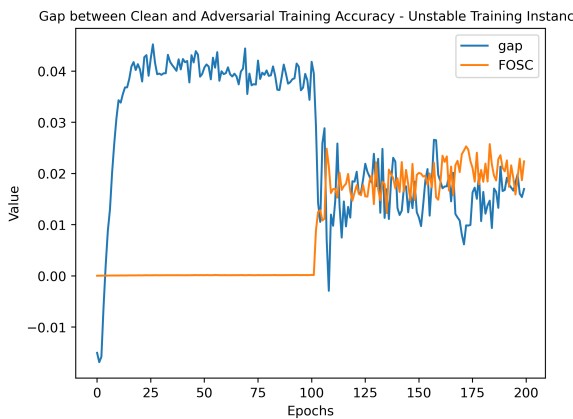

Figure 3: Relationship between FOSC and the gap between the clean training accuracy and adversarial training accuracy. Note that adversarial training accuracy is measured using TPGD. This demonstrates that TPGD, as the training adversary, may cause the model over-fitting to gradient-based attacks, making it difficult for the adversarial example to converge to a good condition.

### 5.1 TPGD CAUSES OVER-FITTING TO GRADIENT-BASED ATTACKS

In Figure 3, by monitoring the gap metric, defined as the difference between the clean training accuracy and the adversarial training accuracy, insights can be gained into the dynamics of robustness overestimation. In instances where the model becomes unstable, as indicated by an increase in the FOSC values, we observe a sudden drop in the gap metric. In some epochs, this gap even temporarily becomes negative, implying that the adversarial training accuracy has surpassed the clean training accuracy. This counterintuitive outcome suggests that the addition of adversarial perturbations through TRADES' PGD (TPGD) somehow aids the model in better classifying the cases, which is clearly abnormal.

As indicated in Equation 1, the inner-maximization step seeks to maximize the adversarial loss by increasing the KL divergence between the clean and adversarial logits. Conversely, the outer minimization step aims to reduce this divergence. This push-and-pull dynamic in logit-logit relations, rather than the typical logit-label relations used in robust methods like PGD-AT (Madry et al., 2018), suggests the model may overfit to the unique characteristics of TPGD perturbations. Further analysis of these dynamics can be found in Appendix F.

The above hypothesis falls in line with the findings from the prior work by Kurakin et al. (2017), which discusses the learnable patterns of simpler attacks like Fast Gradient Sign Method (FGSM) (Goodfellow et al., 2015). However, in this case, the model is not learning any specific patterns; instead, it overfits the perturbations and converges at a point where most gradient-based attackers struggle to cause large logit differences due to obfuscated gradients.

### 5.2 BATCH-LEVEL GRADIENT INFORMATION

To better understand the instability observed during training, we keep track of the weight gradient norm of the full loss (W_grad_norm), the gradient norm of the cross-entropy term (CE_norm), the gradient norm of the KL-divergence term (KL_norm), and the cosine similarity between the gradient directions of the full loss before and after each training step (grad_cosine_similarity). See Appendix G for the detailed definition of these metrics. We examine the changes of these metrics at the batch level in Figure 4, focusing on the epoch where the instability first occurs, namely epoch 102. At this epoch, multiple spikes in W_grad_norm are observed across several batches. Notably, KL_norm also shows spikes in these batches. In contrast, CE_norm remains relatively stable throughout the epochs. These findings support our earlier hypothesis that minimizing the distance between the clean and adversarial logits is a key contributor to the observed instability.

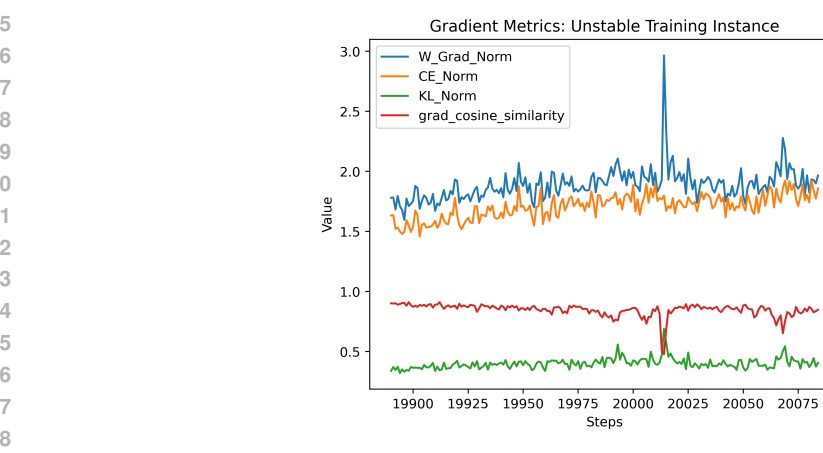

Figure 4: Batch-level gradient metric (detailed definition in Appendix G) of the unstable case at epoch 102, where FOSC starts to rise sharply in Figure 3. (With a batch size of 256, each epoch updates 50,000 / 256 = 195 steps. Therefore, epoch 102 corresponds to steps, ranging from 19,891 to 20,085.) We can see the correlation between W_Grad_Norm, KL_Norm, and grad_cosine_similarity.

Further analysis reveals that the cosine similarity of weight gradients before and after each training step sharply declines at the same points where the KL and total gradient norms spike. This drop in cosine similarity indicates a shift in the direction of the weight gradients, suggesting that the optimization landscape, particularly in the adversarial component, becomes locally rugged. Such ruggedness is a typical trait of non-linear models during adversarial training, where the optimization trajectory tends to be less smooth and more erratic, as noted by Liu et al. (2020).

## 5.3 SELF-HEALING IN TRADES

During our experiments, we observed a phenomenon in some instances of TRADES training that we term "self-healing". This occurs when a model initially shows signs of robustness overestimation but eventually stabilizes, correcting the overestimation without external intervention. Figure 5 illustrates such a case, showing the dynamics of FOSC values, weight gradient norms, and clean training accuracy over the training epochs. As mentioned earlier, instability is marked by spikes in FOSC values, indicating a rugged loss landscape. However, in this example, these spikes are followed by a decline, suggesting a return to a smoother loss landscape and subsequent stability.

A distinctive feature of self-healing instances is a slight decline in clean training accuracy, accompanied by a drop in FOSC to nearly zero within the same epoch. In the following epoch, the weight gradient norm decreases as well, suggesting that the model encountered a challenging batch, triggering the self-healing process and thus leading to a large optimization step, which effectively escapes the problematic local loss landscape that causes obfuscated gradients. Although this step temporarily worsens the model's predictions, it effectively resets the training process to a more stable state.

## 6 SOLVING THE INSTABILITY

Careful hyperparameter tuning and rigorous evaluation may resolve the instability, but this approach requires extensive time spent on trial and error. Therefore, it is essential to design a method within the training process to directly address this issue. Given the goal of resolving the local ruggedness observed in Figure 1, we experimented with Adversarial Weight Perturbation (AWP) (Wu et al., 2020), but it proved ineffective (details in Appendix H). This suggests that previous adversarial training methods, which focused on overall performance, are inadequate for addressing such specific challenges. Consequently, we must leverage the characteristics identified earlier to develop an appropriate solution.

In light of the observed self-healing behavior in Section 5.3, we develop a solution that artificially induces conditions that prompt the model to take larger optimization steps during instability, allow-

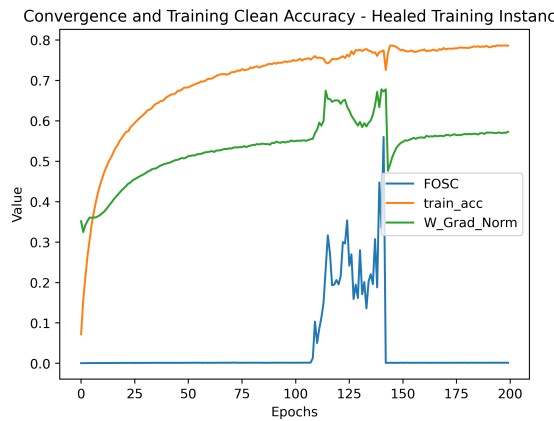

Figure 5: FOSC, training clean accuracy (train_acc), and weight gradient norm (W_grad_norm) of the self-healing case under the same configuration but with a different seed from the training instance in Figure 3. Note that for clearer visualization, we scaled FOSC up by 10 and scaled the W Grad Norm down by 0.3. And it is important to clarify that a decline in clean training accuracy, accompanied by a drop in FOSC to nearly zero, occurs at epoch 142. Subsequently, the weight gradient norm decreases at epoch 143.

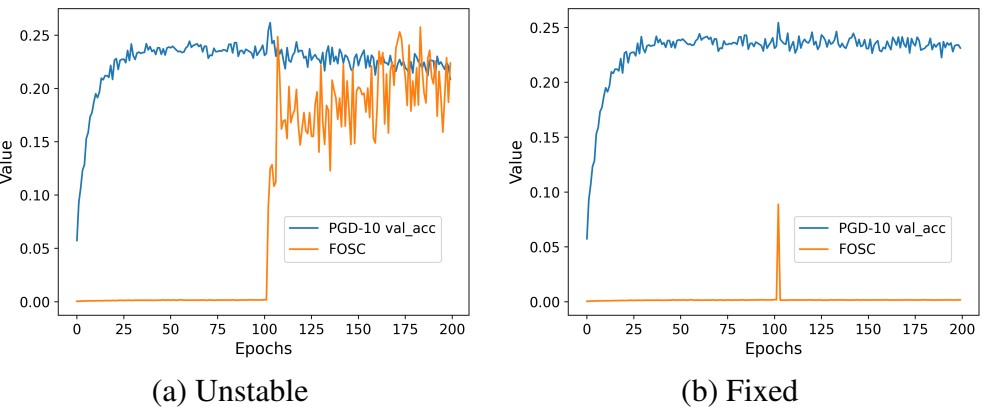

(a) Unstable           (b) Fixed

Figure 6: FOSC and PGD-10 validation accuracy of an Unstable versus Fixed case with the same seed and configuration as Figure 3. Note that for clearer visualization, we scaled FOSC up by 10. Through our method, once the FOSC value is higher than the threshold, we can address the issue in the next epoch.

ing it to escape problematic regions in the loss landscape. This idea is somewhat similar to Ge et al. (2015), where additional noise is introduced to prevent the model from getting stuck at a certain point, caused by the instability.

As outlined in Algorithm 1, we monitor the FOSC values at each training epoch. If the FOSC for a given epoch exceeds a predefined threshold, Gaussian noise is introduced to the images in the first ten batches of the subsequent epoch. This artificial perturbation prompts the model to make substantial adjustments, as the affected batches are likely to produce larger discrepancies between the clean logits and the labels.

Figure 6 presents a case study of an originally unstable training instance with the same seed. Starting from epoch 102, the continually high FOSC value indicates instability. However, simply observing the PGD-10 accuracy makes it difficult to detect this instability. By applying our solution and injecting Gaussian noise into the first ten batches of epoch 103, the FOSC values revert to normal

|  | Clean | PGD-10 | AutoAttack | Gap |
|---|---|---|---|---|
| Regular | $0.5342 \pm 0.0019$ | $0.2440 \pm 0.0006$ | $0.2009 \pm 0.0028$ | $0.0431 \pm 0.0033$ |
| Unstable | $0.5424 \pm 0.0102$ | $0.2763 \pm 0.0133$ | $0.1292 \pm 0.0344$ | $\mathbf{0.1471} \pm 0.0455$ |
| Fixed | $0.5290 \pm 0.0063$ | $0.2450 \pm 0.0023$ | $0.1999 \pm 0.0029$ | $\mathbf{0.0451} \pm 0.0032$ |

Table 3: Performance of different cases under the same configuration, where the Fixed case represents the use of our algorithm and shares the same set of seeds as the Unstable case. Note that Gap represents the difference between the PGD-10 and AutoAttack accuracy. In the Fixed case, there is neither robustness overestimation nor any compromise in performance, which remains consistent with the Regular case.

levels, signifying restored stability. It is worth noting that this solution might still result in a FOSC spike for one epoch, but as long as this checkpoint is not selected (as lines 19-20 of Algorithm 1), our method ensures a good optimization process in all other epochs, similar to the regular case.

Table 3 compares the PGD-10 and AutoAttack accuracies for the unstable and fixed training instances. The results demonstrate that the fixed instance, trained with the proposed algorithm, does not exhibit significant robustness overestimation, unlike the original unstable instance. Notably, our method worsens only a small number of training batches, so it has minimal impact on overall performance when compared to the regular case. Moreover, FOSC is computed alongside adversarial examples during validation, resulting in minimal computational overhead. Finally, rather than continuously adjusting parameter settings to reduce instability or retraining when overestimation occurs—both of which are time-consuming—our approach ensures generalization across different tasks without the risk of severe robustness overestimation.

## 7 LIMITATIONS

This paper focuses primarily on analyzing TRADES, a well-established adversarial training method. Although proper hyperparameter tuning may mitigate some of TRADES' potential issues, we believe these challenges are linked to sampling probability and gradient masking—an issue that should not occur in an adversarial defense method—can still happen. Therefore, its existence should not be overlooked.

Additionally, while many modern defense methods empirically outperform TRADES, it remains a foundational baseline for numerous adversarial training approaches. Researchers continue to build upon TRADES to validate the effectiveness of their own methods. In this sense, TRADES must still maintain a degree of reliability, and this is exactly what we aim to challenge. Moreover, the logit pairing technique used in TRADES raises the question of whether it genuinely enhances robustness or if this is merely a misconception. This issue provides an opportunity for future researchers to further investigate, building on the analysis and solutions we have proposed.

## 8 CONCLUSION AND FUTURE WORK

In conclusion, we have identified the issue of probabilistic robustness overestimation in TRADES, analyzed its root causes, and proposed a potential solution. Hence, we believe that vanilla TRADES should not be fully trusted as a baseline for multi-class classification tasks without applying our solution techniques. Given that TRADES incorporates techniques similar to logit pairing—previously abandoned due to unreliable robustness and gradient masking—we found that TRADES exhibits similar issues. Future research can build upon this correlation by consolidating all these methods to provide a more unified analysis and explanation. Our findings offer a valuable reference for future research in adversarial training methods.

## REPRODUCIBILITY

In Section 3 and Appendix B, G, we explained the experimental methods and settings. Additionally, the source code can be found in the supplementary materials to ensure reproducibility, and we will make the code publicly available after acceptance.

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

## A    SOLUTION ALGORITHM

In Section 6, we explained the solution algorithm to address the instability. Here, we provide a detailed version of the pseudocode.

---

**Algorithm 1** TRADES Training with FOSC Threshold and Gaussian Noise Addition

---

**Require:** $model, train\_loader, val\_loader, epochs, \beta, FOSC_{thresh}$
 1: $best\_adv\_acc \leftarrow 0$
 2: $add\_noise\_batches \leftarrow 0$
 3: $best\_model \leftarrow \phi$
 4: **for** epoch $e$ from 1 to $epochs$ **do**
 5:     **Training Phase**
 6:     **for** each batch $(img, label)$ in $train\_loader$ **do**
 7:         **if** $add\_noise\_batches > 0$ **then**
 8:             $img \leftarrow img + \mathcal{N}(0, 0.1)$
 9:             $add\_noise\_batches \leftarrow add\_noise\_batches - 1$
10:         **end if**
11:         $img\_adv \leftarrow \text{TPGD}(model, img, label)$
12:         $logits\_natural, logits\_adv \leftarrow model(img), model(img\_adv)$
13:         $loss \leftarrow \text{CE}(logits\_natural, label) + \beta \times \text{KL}(logits\_adv, logits\_natural)$
14:     **end for**
15:     **Evaluation Phase**
16:     $clean\_acc, adv\_acc, FOSC \leftarrow \text{evaluate}(model, val\_loader)$
17:     **if** $adv\_acc > best\_adv\_acc$ **and** $FOSC \leq FOSC_{thresh}$ **then**
18:         $best\_model \leftarrow model$
19:     **end if**
20:     **if** $FOSC > FOSC_{thresh}$ **then**
21:         $add\_noise\_batches \leftarrow 10$
22:     **end if**
23: **end for**

---

## B    OTHER EXPERIMENTAL SETTINGS

About other configurations, we follow the similar settings of relevant work (Wu et al., 2024; Rice et al., 2020; Gowal et al., 2020; Pang et al., 2021). We perform adversarial training with a perturbation budget of $\epsilon = 8/255$ under the $l_\infty$-norm. During training, we use a 10-step TPGD (TRADES' PGD) adversary with a step size of $\alpha = 2/255$. The models are trained using the SGD optimizer with Nesterov momentum of 0.9 and a weight decay of 0.0005. For AWP, we choose radius 0.005 as Wu et al. (2024; 2020); Gowal et al. (2020). For CIFAR-10/100 (Krizhevsky et al., 2009), we use 200 total training epochs, and simple data augmentations include $32 \times 32$ random crop with 4-pixel padding and random horizontal flip. As for TinyImageNet-200 (Le & Yang, 2015), we crop the image size to $64 \times 64$ and use 100 training epochs. All experiments were using the ResNet-18 model. Finally, we evaluate the models with PGD-10 at each epoch and select the checkpoint with the highest robust accuracy on the validation set for further experiments.

## C    OTHER DATASETS

The issue of instability in TRADES is not limited to the CIFAR-100 (Krizhevsky et al., 2009) dataset. Instead, when experimenting on CIFAR-10 and Tiny-Imagenet-200 (Le & Yang, 2015), it is clear that robustness overestimation cases can still occur, as demonstrated in Table 4. Generally speaking, we find that instability is more prominent with larger class complexity.

| $\beta$ \ dataset | CIFAR-10 | CIFAR-100 | Tiny-Imagenet-200 |
|---|---|---|---|
| 1 | 1/6 | **10**/10 | **2**/3 |
| 3 | 0/6 | **7**/10 | **3**/3 |
| 6 | 0/6 | 0/10 | 0/3 |

Table 4: Probability of unstable cases w.r.t. $\beta$ and dataset. The batch size is fixed to 256. For each set of parameters, we perform multiple training instances with random seeds to identify groups that may exhibit gradient masking, which are identified when the PGD-10 (white-box) accuracy is significantly higher than the Square Attack (black-box) accuracy. ($\geq 8\%$).

## D   LEARNING RATE SCHEDULING

In all the aforementioned experiments, we did not perform optimizer learning rate scheduling during training in order to better isolate the issue of instability. However, we also conducted experiments to demonstrate that this phenomenon still occurs when the learning rate is decreased throughout the training process. As shown in Table 5, the general trend of smaller $\beta$ values leading to instability remains the same.

|  | Prob. | Clean | PGD-10 | AutoAttack | Gap |
|---|---|---|---|---|---|
| $\beta = 1$ | **3**/3 | $0.6273 \pm 0.0241$ | $0.2880 \pm 0.0179$ | $0.0482 \pm 0.0064$ | $\mathbf{0.2398} \pm 0.0198$ |
| $\beta = 3$ | 0/3 | $0.5860 \pm 0.0120$ | $0.2849 \pm 0.0088$ | $0.2286 \pm 0.0092$ | $0.0563 \pm 0.0151$ |
| $\beta = 6$ | 0/3 | $0.5572 \pm 0.0005$ | $0.2940 \pm 0.0026$ | $0.2440 \pm 0.0012$ | $0.0500 \pm 0.0016$ |

Table 5: Performance of different $\beta$ under the same configuration of learning rate scheduling (CIFAR-100, batch size = 256, initial learning rate $lr_0 = 0.1$, $\gamma = 0.1$, and learning rate denoted $lr_t$ at the $t$-th epoch updated to $lr_{t-1} * \gamma$ at epochs 100 and 150). Note that the first column of Prob. represents the probability of unstable cases and the Gap represents the difference between the PGD-10 and AutoAttack accuracy. The results with different $\beta$ values show that instability still exists and that the general trend of worsening overestimation with smaller $\beta$ values holds.

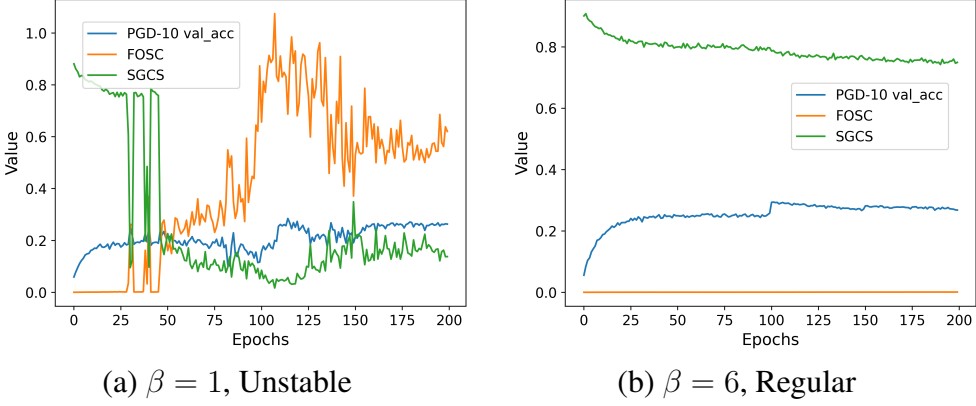

(a) $\beta = 1$, Unstable    (b) $\beta = 6$, Regular

Figure 7: Comparison between the FOSC, SGCS, and PGD-10 validation accuracy under the learning rate scheduling setting. Under the same configuration, the values for the Unstable case with $\beta = 1$ are displayed in (a), while the values for the Regular case with $\beta = 6$ are shown in (b). Note that for clearer visualization, we scaled FOSC up by 5. This aligns with previous findings, and we can observe that even the PGD-10 accuracy exhibits significant fluctuations, indicating severe instability.

## E    RELATION BETWEEN FOSC AND SGCS

In this section, we demonstrate that a non-zero First-Order Stationary Condition (FOSC) value implies a Step-wise Gradient Cosine Similarity (SGCS) less than 1 for a PGD attacker in a toy case where the input $\mathbf{x}$ has dimension of 1. This is done to intuitively show that our hypothesis of poor convergence capability indicating a locally rugged loss landscape is logical, as the SGCS taking a smaller value implies non-aligned steps taken by the PGD attacker.

We work our proof on a PGD-$k$ attacker taking input $\mathbf{x}$ with step-size $\alpha$ and perturbation bound $\epsilon$, assuming $k\alpha > \epsilon$. The perturbation ball is denoted $\mathcal{X} = [\mathbf{x} - \epsilon, \mathbf{x} + \epsilon]$ while the result from the $i$-th perturbation step is $\mathbf{x}^i$.
As demonstrated by Wang et al. (2019), we have the closed-form
$$FOSC(\mathbf{x}^k) = \max_{\delta \in S} \langle \mathbf{x} + \delta - \mathbf{x}^k, \nabla_\mathbf{x}\ell(\mathbf{x}^k) \rangle = \epsilon\|\nabla_\mathbf{x}\ell(\mathbf{x}^k)\|_1 - \langle \mathbf{x}^k - \mathbf{x}^0, \nabla_\mathbf{x}\ell(\mathbf{x}^k) \rangle$$
If $FOSC(\mathbf{x}^k) = 0$, we have that either $\nabla_\mathbf{x}\ell(\mathbf{x}^k) = 0$ or $\mathbf{x}^k - \mathbf{x}^0 = \epsilon\,\text{sign}(\ell(\mathbf{x}^k))$. Namely, either $\mathbf{x}^k$ is a stationary point, or the maximum point for $\ell(x^k)$ is on the boundary of $\mathcal{X}$. This implies that if $FOSC(\mathbf{x}^k) > 0$, $\mathbf{x}^k - \mathbf{x}^0 \neq \epsilon\,\text{sign}(\mathbf{x}^k)$. In other words, $\mathbf{x}^k$ is not on the boundary of $\mathcal{X}$.
Assume that $SGCS(\mathbf{x}^k) = 1$ when $FOSC(\mathbf{x}^k) > 0$. This implies for each pair $i, j$ where $i \neq j$, we have $CosSim(g(\mathbf{x}^i), g(\mathbf{x}^j)) = 1$. Applying the projection in PGD in the one-dimensional case, this means that $\mathbf{x}^k$ can only be $\mathbf{x} \pm \epsilon$, which is the boundary of $\mathcal{X}$. This contradicts $FOSC(\mathbf{x}^k) > 0$.

## F    LOGIT DYNAMICS IN INNER MAXIMIZATION

In order to better understand the effects of maximizing the distance between clean logits and adversarial logits during inner maximization in the TRADES training algorithm, we performed experiments where the original TRADES' PGD (TPGD) was replaced with a standard PGD-10 attacker, which maximizes the distance between adversarial logits and the labels. This approach is similar to the algorithm in Wang et al. (2020), but without the misclassification weight.

As shown in Table 6, this modified algorithm does not lead to robustness overestimation or instability. Thus, we infer that the dynamic of maximizing the difference between clean and adversarial logits is problematic. However, it should be noted that although this training algorithm avoids the issue of instability, it is not an ideal solution due to the significant decrease in clean accuracy.

|          | Clean | PGD-10 | AutoAttack | Gap |
|----------|-------|--------|------------|-----|
| Regular  | $0.5342 \pm 0.0019$ | $0.2440 \pm 0.0006$ | $0.2009 \pm 0.0028$ | $0.0431 \pm 0.0033$ |
| Unstable | $0.5424 \pm 0.0102$ | $0.2763 \pm 0.0133$ | $0.1292 \pm 0.0344$ | $0.1471 \pm 0.0455$ |
| Modified | $\mathbf{0.5026} \pm 0.0057$ | $0.2422 \pm 0.0013$ | $0.1998 \pm 0.0029$ | $\mathbf{0.0424} \pm 0.0022$ |

Table 6: Performance of different cases under the same configuration of learning rate scheduling (CIFAR-100, batch size = 256, learning rate = 0.1). the Gap represents the difference between the PGD-10 and AutoAttack accuracy. The "Modified" case represents training using TRADES outer-minimization and PGD-10 maximization. The results show that the Modified case exhibits almost no instability and has a similar gap value to the Regular case; however, this comes at the cost of clean accuracy.

## G    GRADIENT NORM ANALYSIS

To better understand the behavior of model gradients during the TRADES training process (see Section 5.2), we additionally keep track of the following metrics, using methods similar to Liu et al. (2020):

- **W_Grad_Norm**: The $\ell_2$ norm of the gradient of the full TRADES loss in Equation 1 is denoted as:
$$W\_Grad\_Norm = \left\|\nabla_\theta\left(\mathcal{CE}(f_\theta(\mathbf{x}), y) + \frac{1}{\lambda}\max_{\delta \in S}\mathcal{KL}(f_\theta(\mathbf{x}), f_\theta(\mathbf{x} + \delta))\right)\right\|_2. \quad (4)$$

- **CE_Norm**: The $\ell_2$ norm of the gradient of the first term in TRADES loss is denoted as:

$$CE\_Norm = \|\nabla_\theta \left( \mathcal{CE}(f_\theta(\mathbf{x}), y) \right)\|_2. \tag{5}$$

- **KL_Norm**: The $\ell_2$ norm of the gradient of the second term in TRADES loss is denoted as:

$$KL\_Norm = \left\| \nabla_\theta \left( \frac{1}{\lambda} \max_{\delta \in S} \mathcal{KL}(f_\theta(\mathbf{x}), f_\theta(\mathbf{x} + \delta)) \right) \right\|_2. \tag{6}$$

- **grad_cosine_similarity**: The cosine similarity between the gradient directions of the full loss before and after each training step is:

$$grad\_cosine\_similarity = \frac{\nabla_\theta L^{\text{before}} \cdot \nabla_\theta L^{\text{after}}}{\|\nabla_\theta L^{\text{before}}\|_2 \|\nabla_\theta L^{\text{after}}\|_2}, \tag{7}$$

where $\nabla_\theta L^{\text{before}}$ and $\nabla_\theta L^{\text{after}}$ represent the gradients of the full TRADES loss, as defined in Equation 4, before and after a training step, respectively.

All four norm metrics are calculated in each training step. If the value of a metric is referenced at the epoch level, the average value across training steps is taken.

## H  ADVERSARIAL WEIGHT PERTURBATION

Hypothesizing that local ruggedness (as shown in Figure 1) causes the PGD-10 attacker to fail in finding effective adversarial examples, we attempted to use a sharpness-aware technique, specifically Adversarial Weight Perturbation (AWP) (Wu et al., 2020) as a solution.

However, we found that adding this technique does not eliminate instability when used with the TRADES algorithm. In fact, as shown in Table 7, all training instances displayed robustness overestimation and instability. We hypothesize that this is due to small rugged areas that are not smoothed out when flattening the landscape in general.

| | Prob. | Clean | PGD-10 | AutoAttack | Gap |
|---|---|---|---|---|---|
| $\beta = 1$ | **10**/10 | $0.6023 \pm 0.0305$ | $0.2880 \pm 0.0145$ | $0.1005 \pm 0.0148$ | $\mathbf{0.1875} \pm 0.0238$ |
| $\beta = 3$ | **10**/10 | $0.5707 \pm 0.0013$ | $0.3180 \pm 0.0187$ | $0.1623 \pm 0.0320$ | $\mathbf{0.1557} \pm 0.0496$ |

Table 7: Performance of different $\beta$ under the same configuration of AWP (CIFAR-100, batch size = 256, learning rate = 0.1). Note that the first column of Prob. represents the probability of unstable cases and the Gap represents the difference between the PGD-10 and AutoAttack accuracy. We can see that applying AWP techniques does not solve the issue of robustness overestimation.

