# OpenReview forum: "Adversarial Robustness Overestimation and Instability in TRADES"
_ICLR.cc/2025/Conference — ICLR 2025 Conference Withdrawn Submission_

### Official Review · Reviewer_arDT · 2024-10-17

**Soundness:** 3
**Presentation:** 2
**Contribution:** 2
**Rating:** 3
**Confidence:** 4

**Summary:**

This paper reveals the overestimation of adversarial robustness (performance gap between AA and PGD) in the TRADES method. The authors attribute this discrepancy to the phenomenon of gradient masking and introduce an approach that involves Gaussian noise to address the identified instabilities.

**Strengths:**

1. TRADES is a prominent, trusted, and widely used AT method, revealing its potential pseudo-robustness is novel and important.
2. The motivation and method are clear, and the visualization is helpful.

**Weaknesses:**

1. While it is notable that TRADES exhibits gradient masking, it is more important and interesting to gain a deeper understanding of why TRADES and logit paired methods are prone to gradient masking, while methods such as PGD are not. What is fundamentally different between their mechanisms?

2. The authors discuss a range of abnormal phenomena associated with unstable TRADES, including distorted loss landscape [1], the negative gap between clean and adversarial training accuracy [2],  weight gradient norm [3], and self-healing [4]. However, these abnormal phenomena have been well-studied in the field of "catastrophic overfitting" within FGSM-AT, suggesting that these findings might not be very novel.

3. Why do not just simply add random initialization to all images just like the PGD dose, but using FOSC value as an indicator? The specific benefits of utilizing FOSC value over simpler methods warrant further clarification.

4. The model occurrence gradient masking typically shows an abnormally higher PGD accuracy (as shown in Tab. 1), but why the regular and unstable model's PGD-10 accuracy is almost the same (as shown in Fig. 2)?

[1] Kim, H., Lee, W., & Lee, J. (2021, May). Understanding catastrophic overfitting in single-step adversarial training. In Proceedings of the AAAI Conference on Artificial Intelligence (Vol. 35, No. 9, pp. 8119-8127).\
[2] Lin, R., Yu, C., & Liu, T. (2024). Eliminating catastrophic overfitting via abnormal adversarial examples regularization. Advances in Neural Information Processing Systems, 36.\
[3] Andriushchenko, M., & Flammarion, N. (2020). Understanding and improving fast adversarial training. Advances in Neural Information Processing Systems, 33, 16048-16059.\
[4] Li, B., Wang, S., Jana, S., & Carin, L. (2020). Towards understanding fast adversarial training. arXiv preprint arXiv:2006.03089.

**Questions:**

See weakness.

---

### Official Review · Reviewer_4M9q · 2024-10-22

**Soundness:** 1
**Presentation:** 3
**Contribution:** 1
**Rating:** 1
**Confidence:** 5

**Summary:**

This work shows that when changing the batch size, regularization parameter $\beta$ and number of classes, the TRADES training can result in probabilistic robustness overestimation, where the gap between the AutoAttack and PGD-10 validation accuracies significantly grows in a few steps during training. Authors observe that probabilistic robustness overestimation occurs simultaneously with gradient masking. Then, as a solution to probabilistic robustness overestimation, authors propose injecting noise to samples before the PGD-10 attack during training when gradient masking is detected.

**Strengths:**

- The paper is well written and easy to follow.
- The probabilistic robustness overestimation problem is well displayed and the co-location with gradient masking is clear.

**Weaknesses:**

I mainly have two concerns:

- **The lack of discussion about the relationship with Catastrophic Overfitting (CO) [1,2]**

Many of the characteristics of probabilistic robustness overestimation have been observed in CO, as the gap between single-step adversarial accuracy and multi-step adversarial accuracy, the sudden increase of such gap, the sensitivity to hyper-parameters and the distortion of the loss landscape. The metrics used to detect probabilistic robustness overestimation (SGCS) are practically the same as to detect CO, see Gradient Misalignment in [2]. Even the solution proposed by authors (adding noise prior the attack), has been already considered to avoid CO [1,3] with arguable effectiveness [4]. In the paper, no reference to CO is provided, even though the problem and solutions are similar.

- **The probabilistic robustness overestimation is an artifact coming from wrongly chosen hyper-parameters**

Authors choose $\beta=3$ in the majority of their experiments. Nevertheless, the original TRADES paper and later reproductions suggest using $\beta=6$ [5,6]. Authors argue in section 3.1 “The above setting is relatively reasonable and similar to [5]…”. But then, in Table 4, authors show that when using the original $\beta=6$, the probabilistic robustness overestimation problem disappears. Authors conclude their work in lines 474-476 with the statement “we believe that vanilla TRADES should not be fully trusted as a baseline for multi-class classification tasks without applying our solution techniques”. This conclusion is misleading as the problem only occurs when $\beta$ is changed. TRADES is considered in the literature as a reliable method to increase the robustness of classifiers [6].

Moreover, there is no point in changing $\beta$ from $6$ to $3$ as no benefits appear. In the case of CO, we would like to reduce the number of PGD steps in training as much as possible in order to reduce computation. Therefore, it makes sense to propose solutions to CO [2,3,4]. In the case of probabilistic robustness overestimation, the problem can be solved by simply selecting the appropriate hyperparameters.

For these reasons, I cannot propose the paper for acceptance.

**References:**

[1] Wong et al., Fast is better than free: Revisiting adversarial training, ICLR 2020

[2] Andriushchenko and Flammarion, Understanding and Improving Fast Adversarial Training. NeurIPS, 2020.

[3] de Jorge et al., Make Some Noise: Reliable and Efficient Single-Step Adversarial Training. NeurIPS, 2022.

[4] Abad Rocamora et al., Efficient local linearity regularization to avoid Catastrophic Overfitting. ICLR 2024.

[5] Zhang et al., Theoretically Principled Trade-off between Robustness and Accuracy., ICML 2019.

[6] Xu et al., Exploring and Exploiting Decision Boundary Dynamics for Adversarial Robustness, ICLR 2023

**Questions:**

- How does probabilistic robustness overestimation relate to Catastrophic Overfitting?

- In the original TRADES paper and other reproductions they recommend using $\beta=6$, why do you use $\beta=3$?

---

### Official Review · Reviewer_6VfH · 2024-11-05

**Soundness:** 2
**Presentation:** 2
**Contribution:** 1
**Rating:** 3
**Confidence:** 4

**Summary:**

The paper focuses on investigating TRADES which is a well known adversarial training method.

The paper identifies hyper-parameter settings in which training with TRADES can result in non-robust models with obfuscated gradients. This is evidenced by having a higher accuracy under a white-box attack (PGD10) compared to a black-box attack (Square, as part of the well known AutoAttack attack bundle). The paper shows that only in a subset of training runs the classifiers are not robust, which seems to depend solely on a random seed.

The paper proceeds to collect a few metrics from the literature (e.g., FOSC, SGCS) and shows how they can be used to identify when a training run is "unstable" -- i.e. if training is continued this will result in obfuscated gradients.

The paper proposes a simple method to help combat such metrics, by adding noise to the training images when a selected metric crosses a pre-specified threshold.

**Strengths:**

The paper starts of interesting -- by posing the question: Does TRADES truly result in robust classifiers? And then proceeding to find configurations where TRADES might not be.

It is also interesting to find cheap metrics which correlate with a robustness gap.

**Weaknesses:**

In modern deep learning papers, which are empirical in nature, tuning of models is required. This is literally bread and butter for any ML practitioner.

The paper states that their motivation is to be invariant to hyper-parameters in TRADES, and just fix TRADES when the hyper-parameters are off. But, the paper introduces (at least) two extra hyper-parameters -- the threshold on FOSC and the amount of noise to add (set at 0.1 perhaps).

It seems that it would have been easier to just tune the hyper-parameters. E.g., the TRADES beta parameter, learning rate, batch size, as is normally done.

Or, better yet, just to download a codebase where everything was tuned already. Also, note that only a very small model has been used (ResNet18), which is not close to a SOTA model.

So, the main takeaway of the paper -- that an untuned set of hyper-parameters results in a non-robust classifier, is not surprising at all. Too high of a regularizer weight (beta), of course, breaks model learning.

I suggest the authors consider proceeding by visiting, e.g., RobustBench, downloading some codebases, and re-running some experiments there. There is limited utility in trying to draw empirical conclusions from un-tuned models.


Also, the paper should clarify what the random seed refers to? The model initialization? The perturbation initialization? The dataset reading/shuffling seed? All of them?


Other comments:
- At a high-level, the paper is overly verbose and repetitive.

- The sample size is way too small for the claims made -- 6 models for CIFAR10, 10 for CIFAR-100, 3 for Tiny-ImageNet (see Table 4). I recommend the authors re-run this analysis.

- "sign" seems to perhaps mean l2 normalize in the paper, maybe? The paper should be clarified -- see for example line 147 and the caption in figure 1.

- "Self-healing" is just a gradient step on a high loss batch, that moves the model parameters in a different part of the optimisation landscape. This is in line with the observed instability that seems to be dependent on the model initialization (assuming that is what the random seed controls).

- The proposed healing mechanism of adding noise, has been explored at lengths in the adversarial robustness literature, but the paper makes no reference to "Certified Adversarial Robustness via Randomized Smoothing" (Cohen, Rosenfeld, Kolter '19).

- The paper makes an claim in section 6, on "previous adversarial training methods [...] are inadequate for addressing such specific challenges". Given that only one other method was attempted as a fix, this feels unsubstantiated.

**Questions:**

See above.

---

### Author Response · Authors · 2024-11-25

Dear Reviewers,

We are grateful for your feedback.  As previously noted, there are several areas that require improvement in this work. Consequently, we have elected to withdraw it from consideration for the time being.

We would nevertheless like to reply to certain issues that have been raised and would appreciate the opportunity to engage in further discussion and receive feedback.

1. In regard to the selection of the beta value:
    - We acknowledge that the original work suggests a beta value of 6 and recognize the difficulty in reproducing robust overestimation with those settings.
    - We maintain that an examination of lower beta values is advantageous for two reasons:
        a. The objective is to establish a benchmark method that is relatively stable in general and not significantly reliant on hyperparameter tuning. If modifying a regularizer weight by a factor of 0.5 results in overestimation, this is a notable consequence.
        b. There is a tradeoff when utilising a high value of beta (e.g., 6). Given the characteristics of the loss function, an emphasis on robustness (as reflected in the weight assigned to the KL term) necessarily entails a compromise in terms of the priority accorded to a clean loss.
    - Additionally, some reviewers have indicated that: A regularizer weight that is "too high" will cause the algorithm to malfunction. This is, to our understanding, contrary to the actual scenario. It is noteworthy that a "lower" regularizer weight results in overestimation.  It would be reasonable to assume that a lower weight would result in a lower PGD accuracy, rather than the opposite.

2. FOSC threshold and Gaussian noise amount:
    - Some reviewers have suggested that, rather than introducing an FOSC threshold, it may be more straightforward to simply tune the existing hyperparameters.
    - However, we hypothesize that the spikes in FOSC versus the normal state can be used as a binary indicator of whether gradient masking is occurring. Additionally, in a dataset with fixed pixel value ranges, the Gaussian noise amount could be fixed. Therefore, this solution requires less explicit tuning.
    - The reason we do not add noise to every image is that we want to maintain as much useful information as possible, and only "sacrifice" a small amount of data when a fix to evident gradient masking is necessary.

3. Relevant Literature:
We extend our apologies for having overlooked the branch of work on catastrophic overfitting. We will conduct further research into this area before considering future work on this paper.

---

### Note · Authors · 2024-11-25

I have read and agree with the venue's withdrawal policy on behalf of myself and my co-authors.